# Unified theoretical framework for black carbon mixing state allows greater accuracy of climate effect estimation

Jiandong Wang [1,2,14] ✉, Jiaping Wang [3,4,14] ✉, Runlong Cai[5], Chao Liu[1,2], Jingkun Jiang [6], Wei Nie [3,4], Jinbo Wang[3], Nobuhiro Moteki[7], Rahul A. Zaveri [8], Xin Huang [3], Nan Ma[9], Ganzhen Chen[2], Zilin Wang [3], Yuzhi Jin[2], Jing Cai[5], Yuxuan Zhang [3,4], Xuguang Chi[3,4], Bruna A. Holanda [10,13], Jia Xing [6], Tengyu Liu [3,4], Ximeng Qi[3,4], Qiaoqiao Wang[9], Christopher Pöhlker [10], Hang Su [10], Yafang Cheng [10], Shuxiao Wang [6], Jiming Hao[6], Meinrat O. Andreae [10,11,12] & Aijun Ding [3,4] ✉

Black carbon (BC) plays an important role in the climate system because of its strong warming effect, yet the magnitude of this effect is highly uncertain owing to the complex mixing state of aerosols. Here we build a unified theoretical framework to describe BC's mixing states, linking dynamic processes to BC coating thickness distribution, and show its self-similarity for sites in diverse environments. The size distribution of BC-containing particles is found to follow a universal law and is independent of BC core size. A new mixing state module is established based on this finding and successfully applied in global and regional models, which increases the accuracy of aerosol climate effect estimations. Our theoretical framework links observations with model simulations in both mixing state description and light absorption quantification.

As a ubiquitous aerosol component, black carbon (BC) is a major contributor to global warming owing to its strong light absorption[1–7], which strongly depends on its mixing state[2,8–15]. Undergoing multiple atmospheric processes, freshly emitted BC can be internally mixed with other aerosol components (i.e., BC coating) and its light absorption is enhanced by the "lensing effect"[1,16–19]. In the real atmosphere, the mixing state of BC-containing particles is a complex property

related to several processes, including condensation, coagulation, and deposition. Many studies have characterized mixing state properties from different aspects based on field observations[8,9,20–26]. However, the overall effect of these dynamic processes on BC mixing state is not well understood. Moreover, the complexity and diversity of BC mixing states in the real atmosphere cannot be represented in global climate models, and therefore these models generally use simplified schemes,

[1]Collaborative Innovation Center on Forecast and Evaluation of Meteorological Disasters, Nanjing University of Information Science and Technology, 210044 Nanjing, China. [2]China Meteorological Administration Aerosol-Cloud-Precipitation Key Laboratory, School of Atmospheric Physics, Nanjing University of Information Science and Technology, 210044 Nanjing, China. [3]Joint International Research Laboratory of Atmospheric and Earth System Sciences, School of Atmospheric Sciences, Nanjing University, 210023 Nanjing, China. [4]National Observation and Research Station for Atmospheric Processes and Environmental Change in Yangtze River Delta, 210023 Nanjing, China. [5]Institute for Atmospheric and Earth System Research / Physics, Faculty of Science, University of Helsinki, 00014 Helsinki, Finland. [6]State Key Joint Laboratory of Environment Simulation and Pollution Control, School of Environment, Tsinghua University, 100084 Beijing, China. [7]Department of Earth and Planetary Science, Graduate School of Science, The University of Tokyo, Tokyo 113-0033, Japan. [8]Atmospheric Sciences & Global Change Division, Pacific Northwest National Laboratory, Richland, WA 99352, USA. [9]Institute for Environmental and Climate Research, Jinan University, 511443 Guangzhou, China. [10]Max Planck Institute for Chemistry, 55128 Mainz, Germany. [11]Scripps Institution of Oceanography, University of California San Diego, La Jolla, CA 92093, USA. [12]Department of Geology and Geophysics, King Saud University, 11451 Riyadh, Saudi Arabia. [13]Present address: Hessian Agency for Nature Conservation, Environment and Geology, 65203 Wiesbaden, Germany. [14]These authors contributed equally: Jiandong Wang, Jiaping Wang. ✉e-mail: jiandong.wang@nuist.edu.cn; wangjp@nju.edu.cn; dingaj@nju.edu.cn

assuming either an internal or external mixture of aerosols[2,27], leading to a wide range of estimated BC mass absorption cross-sections (MAC, a typical indicator of BC light absorption ability) from 3.1 to 18.0 m²/g (at 550 nm) on global average[27,28]. Therefore, a precise description of BC mixing state becomes the determinant factor of model performance when estimating BC optical properties and radiative forcing.

In this study, we built a theoretical framework linking dynamic processes to BC coating thickness distribution and discovered the self-similarity of BC mixing states, which was verified in eight different observation sites globally. The size distribution of BC-containing particles is found to follow a universal law. This self-similarity allows us to greatly simplify the characterization of BC mixing states in both model simulations and field observations. A new mixing state scheme was established for model simulation, which can precisely represent the BC mixing state. Model estimated BC absorption and radiative forcing is substantially reduced, which fits well with available field observation results. Our study links observations with model simulations in both mixing state and light absorption.

## Results

### A universal law of BC mixing state

To characterize the BC mixing state, which is controlled by complex atmospheric processes, we performed a theoretical derivation considering the main physical processes affecting BC in the atmosphere. We discovered that the size distribution of BC-containing particles follows a universal law and is independent of BC core size. Figure 1 provides a conceptional scheme describing the main physical processes and evolution of aerosols in the atmosphere. We assume a monodisperse aerosol population (consisting of BC cores only) emitted into the atmosphere at time zero with diameter $D_c$ and number concentration $n(D_c)$. After being emitted into the atmosphere, the particles experience both growth and deposition progresses continuously, which form a steady state[29], that is, the size distribution of BC-containing particles is approximately steady (although the overall mass concentration may change). The growth of BC-containing particles via condensation and coagulation results in an enlarged particle size, whose change as a function of time is

represented by the growth rate (GR).

$$\frac{d(D_p)}{dt} = GR \tag{1}$$

The time evolution of the diameter of BC-containing particles ($D_p$) can be integrated to give

$$D_p = GR \cdot t + D_c \tag{2}$$

At the same time, BC particles are removed by deposition process with the rate of Dep. The number concentration of particles at $D_p$, i.e., $n(D_p)$, decays due to deposition process is

$$dn(D_p) = -Dep \cdot n(D_p)dt \tag{3}$$

Then, the time evolution of $n(D_p)$ can be integrated as

$$n(D_p) = n(D_c) \cdot e^{-Dep \cdot t} \tag{4}$$

Based on the steady-state approximation, Eqs. 2 and 4 can be combined and time term $t$ can be eliminated. Taking the logarithm on both sides, we obtain the following equation:

$$\ln(n(D_p)) = \ln(n(D_c)) - \frac{Dep}{GR}(D_p - D_c) \tag{5}$$

The slope $k = \frac{Dep}{GR}$ is determined by the deposition rate and the growth rate, and the intercept is determined by $n(D_c)$. Note that we adopted a simplified derivation in the above theoretical analysis for better understanding. A more rigorous theoretical derivation as well as the interpretation of the dependency of GR and Dep on $D_p$ and time can be found in the Supplementary Information (SI).

Equation 5 demonstrates that for different particle sizes, $\ln(n(D_p))$ and $D_p - D_c$ ($\Delta D_p$, defined as coating thickness) are in a linear relationship (Fig. 1). The average coating thickness can be derived as $1/k$ (the detailed formula is shown as Eq. 7 in the "Methods" section). Furthermore, we find that the slope $k$ is independent of $D_c$, indicating the self-similarity of BC size distributions, that is, BC-containing aerosol with different core sizes should have similar distributions of coating thickness. Such self-similarity allows us to greatly simplify the description of BC mixing states since BC coating thickness can be fully described with known slope $k$ and the BC core number-size distribution.

We verified our theoretical model by field observations of BC size using the single particle soot photometer (SP2, Droplet Measurement Technologies, USA), which is a well-recognized instrument to measure BC mixing state) from eight sites covering different environments globally[24,30,31]. As presented in Fig. 2, the BC size distribution follows an exponential law at all sites despite their different regions and properties (e.g., urban, regional background), which agrees well with our theoretical model. The slope $k$ of linear regression, ranging from 0.008 to 0.020, provides a useful parameter to quantify BC size distribution and its absorption enhancement.

The $D_p$ distributions with different BC core sizes from SP2 measurements at four sites are presented in Fig. 3. We selected four $D_c$ bins (110–120 nm, 120–130 nm, 130–140 nm, and 140–150 nm) and calculated the average $D_p$ distribution in each bin. The result shows that the BC size distributions with different $D_c$ have approximately the same slope of $\ln(n(D_p))$-$D_p$. This phenomenon can be observed at all four sites, validating our theoretical model presented as Eq. 5, i.e., the shape of the BC size distribution is independent of $D_c$. The presence of the same pattern in Nanjing (suburban), Maqu (remote

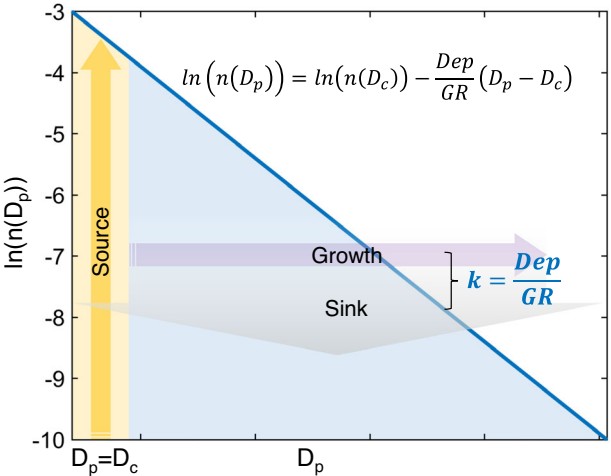

**Fig. 1 | Similarity of black carbon (BC) size distribution controlled by combined effects of growth and sink.** The yellow shaded area and arrow show the emission source of BC where BC core size ($D_c$) is approximately equal to the diameter of BC-containing particles ($D_p$). The blue shaded area and line represent the size distribution of $D_p$ controlled by growth (purple arrow) and sink (gray arrow) processes. The slope of the blue line equals to $-\frac{Dep}{GR}$ in a $\ln(n(D_p))$–$D_p$ coordinate system.

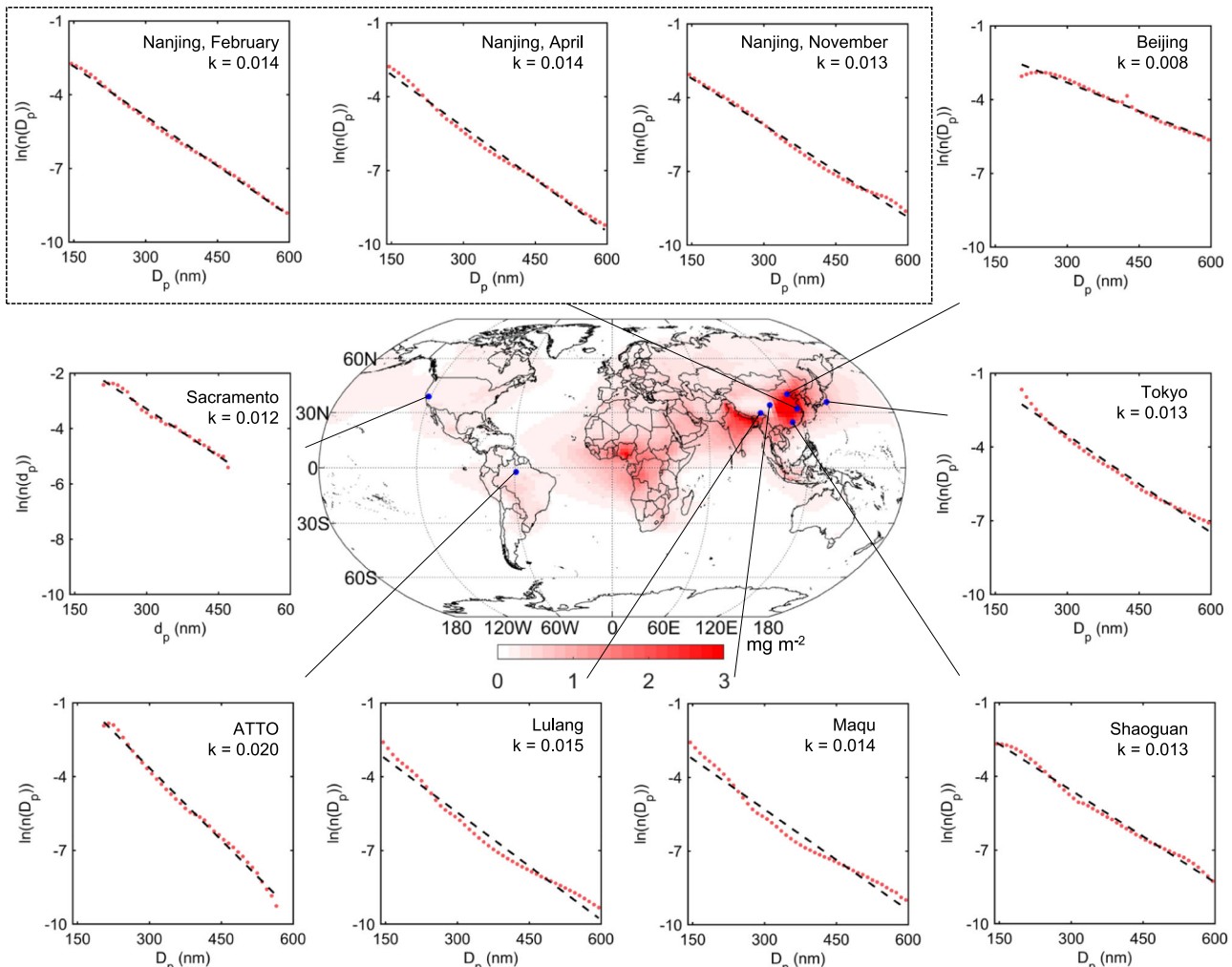

**Fig. 2 | Exponent distribution of black carbon (BC) size distributions from field measurements using single particle soot photometers (SP2) at different sites.** The red dots and black lines represent the normalized size distribution of the diameter of BC-containing particles ($D_p$) and linear regression, respectively. Observations in Nanjing were conducted in different seasons. Data in Lulang and Maqu were observed over the Tibetan Plateau in spring 2021. Observations in Shaoguan, Beijing[31], Tokyo[30], Sacramento[24], and Amazon Tall Tower Observatory (ATTO) were performed in December 2020, November 2014, August 2012, June 2010, and October 2019, respectively. The map shows simulated BC radiative forcing at top of atmosphere (TOA).

background), Tokyo (urban), and at the Amazon Tall Tower Observatory site (affected by biomass burning) further indicates that the self-similarity of BC size distribution is ubiquitous in the real atmosphere.

**Improved estimation of BC absorption and radiative effect**
When evaluating BC absorption and radiative effect, simply assuming either internal or external mixture of aerosols may induce large discrepancies and uncertainty. Also, it is difficult to provide an accurate description of mixing state due to its great complexity. Based on our theory, the description of BC mixing state can be greatly simplified, which is applicable to the light absorption calculation in both climate models and chemical transport models. The absorption enhancement factor, $E_{abs}$, is the ratio between the aerosol absorption coefficients before and after removal of coating, which is a widely used parameter to represent BC light absorption amplification. As shown in Fig. S3, the relationship between $E_{abs}$ and $\Delta D_p$ is approximately linear when $\Delta D_p$ is small. Therefore, the BC coating thickness distribution can be replaced by a monodisperse coating thickness, $1/k$, when calculating the black carbon absorption (a detailed demonstration can be found in the "Methods" section). Light absorption coefficients over all $D_p$ and with mean $D_p$ show good agreement (Fig. S4), which further validates this

simplification. Hence, this approximation can be applied directly in global and regional models for optical estimation.

Based on the above findings, we established a new mixing state module and applied it in a global climate model (CESM-CAM6) and a chemical transport model (WRF-chem) as examples. Model simulations of $E_{abs}$ and BC direct radiative forcing (DRF$_{BC}$) were performed using these two models alternatively with our module and the conventional assumption of mixing state (Figs. 4 and 5). Comparing the simulated $E_{abs}$ with observational data shown in Fig. 4, the simulated $E_{abs}$ using the conventional assumption of mixing state in both CESM-CAM6 and WRF-Chem (blue squares) are significantly higher than observations (1.0 to 1.7, shown as black squares). CESM-CAM6 uses the volume mixing assumption for $E_{abs}$ calculation and the simulated result is higher than 2.5. WRF-Chem has two types of mixing state assumption, which are volume mixing and core-shell mixing. However, the simulated $E_{abs}$ using both types of assumptions ranges from 2.0 to 2.5, which is also nearly twice that of observational results. Using the new module, the calculated $E_{abs}$ in CESM-CAM6 and WRF-Chem are 1.4 (1.3–1.6) and 1.4 (1.3–1.7), respectively, which agrees well with observational data, demonstrating the good performance of our mixing state description in the quantification of $E_{abs}$.

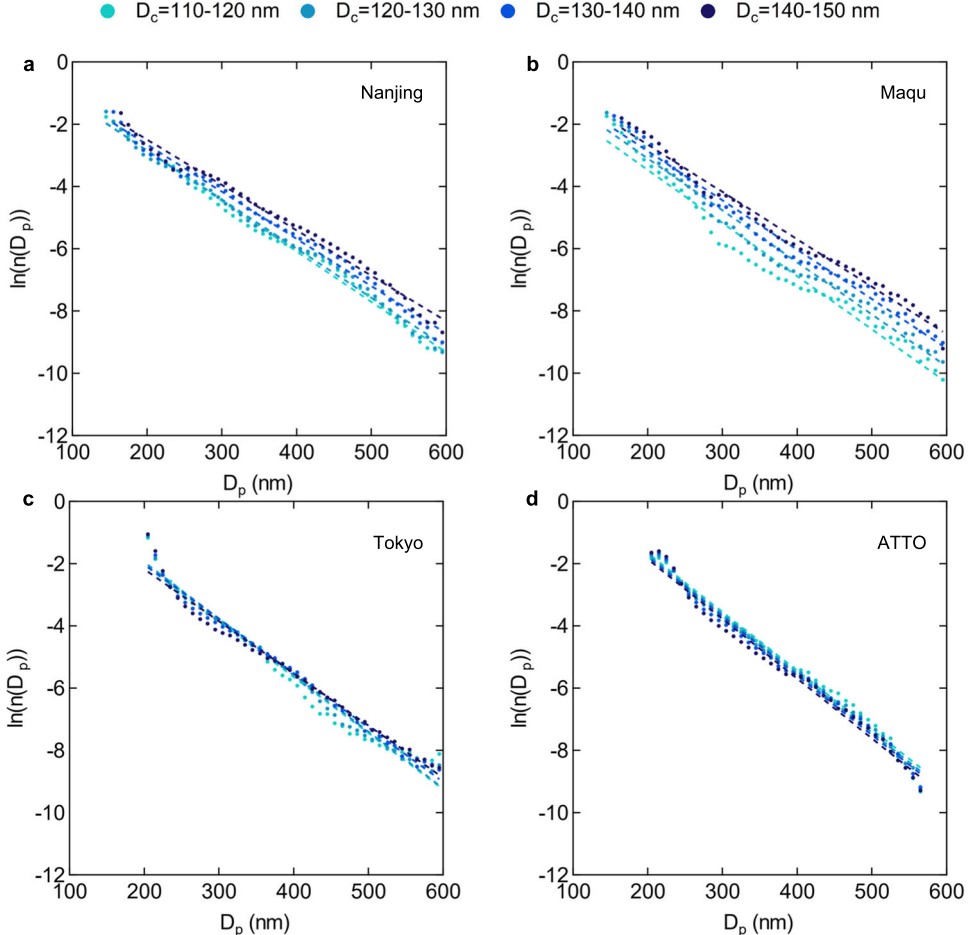

**Fig. 3 | Black carbon (BC) size distributions with different $D_c$ ranges at four sites.** Single particle soot photometers (SP2) observational data from **a** Nanjing (April), **b** Maqu, **c** Tokyo, and **d** Amazon Tall Tower Observatory (ATTO) sites is used. Circles are the diameter of BC-containing particles ($D_p$) distributions with four selected BC core size ($D_c$) ranges. Dashed lines represent the linear regression of each distribution. $n(D_p)$ with each $D_c$ range normalized.

Figure 5 presents simulated $DRF_{BC}$ using the new scheme and the conventional scheme in CESM-CAM6. We selected four typical regions (Europe, North America, South America, and Asia) to perform statistical analyses. Calculated $DRF_{BC}$ using our scheme is greatly reduced in all four regions (40–50%) compared with adopting the conventional scheme. This result is in agreement with the recently-found overestimation of radiative warming by BC in global climate models, largely due to the treatment of aerosol mixing state[32]. Figure 5 demonstrates that the new scheme based on our theoretical framework can efficiently resolve the existing overestimation of model-simulated radiative warming by BC.

## Discussion

We built a unified theoretical framework based on a steady-state assumption to describe the complex mixing state of BC and find its self-similarity confirmed across a wide range of field observation sites. This universal law is the result of a balance of multiple atmospheric processes. Our findings link the representation of particle diameter (from field observations) and dynamic parameters (generally used in models), making observational data applicable in model simulations. Moreover, this unified theoretical framework reduces the dimension of mixing state descriptions. The mixing state module established in this study can be embedded in various types of atmospheric models and efficiently improves the accuracy of aerosol climate effect estimations without increasing computational complexity. We find that

BC absorption enhancement and warming effect are much lower than current estimates.

## Methods

### Light absorption enhancement

We preformed the optical calculation using the core-shell Mie method. We used a lognormal distribution of $D_c$. The geometric standard deviation was set to 1.8. The mean diameter of $D_c$ was 70 nm and the wavelength was 550 nm. The refractive indices (RI) of the BC and scattering components were set to 1.85 + 0.7i according to Bond et al.[1] and 1.53 + 0i according to Pitchford et al.[2]. To derive the response of MAC to $\Delta D_p$ (Fig. S3), $\Delta D_p$ from 10 nm to 200 nm with 10 nm intervals was adopted to calculate the mass absorption coefficient.

For the optical calculations in Fig. S4, we used the integral method ($\Delta D_p$ varied from 1 nm to 1000 nm with 1 nm interval) and the $k$-value method to calculate absorption coefficients. The $D_p$ size distribution represented as Eq. 5 was adopted and $k$ was set to 0.014. The setting of RI, wavelength, and $D_c$ size distribution were same as in Fig. S3.

As shown in Fig. S3, the relationship between $E_{abs}$ and $\Delta D_p$ is approximately linear when $\Delta D_p$ is small. Therefore, $E_{abs}$ could be represented as Eq. 6.

$$E_{abs}(D_c, \Delta D_p) = \alpha(D_c)\Delta D_p \tag{6}$$

where $\alpha(D_c)$ is the linear coefficient of $E_{abs}$ and $\Delta D_p$. With the known calculation formula of $\overline{D_p}$, we can derive that

$$\overline{\Delta D_p} = \frac{\int_{D_p=D_c}^{\infty} D_p \cdot n(D_p) \cdot d(D_p)}{\int_{D_p=D_c}^{\infty} n(D_p) \cdot d(D_p)} = \frac{1}{k} \qquad (7)$$

If Eq. 6 is integrated over all $D_p$, the average $E_{abs}$ with given $D_c$ is found to be

$$\begin{aligned} E_{abs}(D_c) &= \frac{\int_{D_p=D_c}^{\infty} c_{abs\text{-}external}(D_c) \cdot E_{abs}(D_c, \Delta D_p) \cdot n(D_c, D_p) \cdot d(D_p)}{\int_{D_p=D_c}^{\infty} c_{abs\text{-}external}(D_c) \cdot n(D_c, D_p) \cdot d(D_p)} \\ &= \alpha(D_c) \cdot \overline{\Delta D_p} \\ &= \alpha(D_c) \cdot \frac{1}{k} \\ &= E_{abs}(D_c, \overline{\Delta D_p}) \end{aligned} \qquad (8)$$

where $c_{abs\text{-}external}$ represents the light absorption coefficients of BC core. Equations 7 and 8 show that $1/k$ plays a similar role with coating thickness.

### Field observations and site descriptions

Observational data of BC mixing states was collected from different sites, including Nanjing (a regional background site in the Yangtze River Delta region in China), Beijing (an urban site in the capital of China), Shaoguan (a regional background site in the Pearl River Delta region in China), the Tibetan Plateau (including three sites with different features), Japan (Tokyo), and the United States (Sacramento, influenced by biomass burning).

Field measurements in Nanjing were performed at the Station for Observing Regional Processes of the Earth system (SORPES, 118°57′10″ E, 32°07′14″ N; 40 m a.s.l.), a regional background station in the western YRD region. A detailed description of SORPES can be found in previous studies[33,34]. Due to its geographical position, this observation platform is ideal to capture the transport from megacities in the YRD region and North China Plain. Observational data from February 2020, April 2020, and December 2021 was used in this study.

Observations in Lulang and Maqu were made over the Tibetan Plateau (TP) from April to July 2021. The Lulang site is located on the southeastern part of the TP with few traffic emissions nearby. The measurement period in Lulang was from 1 April to 25 May 2021. Maqu can be considered as a background site over the TP. The measurement

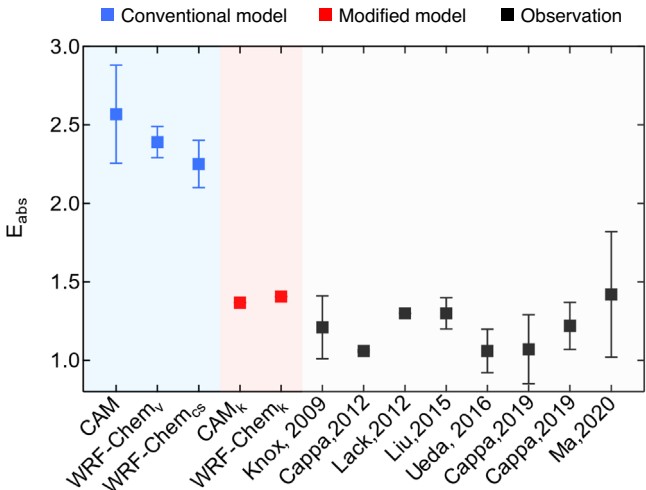

**Fig. 4 | Model simulated black carbon (BC) absorption enhancement using the new scheme in this study (red squares) and the conventional scheme (blue squares), compared with observations.** WRF-Chem$_v$ and WRF-Chem$_{cs}$ stand for WRF-Chem simulations with volume mixing and core-shell mode, respectively. The black squares with error bars (standard deviation) are $E_{abs}$ observed using the thermodenuder (TD) method at different sites obtained from previous studies[9,13,48–52]. There are two exceptions, which are Knox et al.[48] and Ueda et al.[50]. The error bars in Knox et al.[48] represent the $E_{abs}$ of aerosol with different age categories and the error bars in Ueda et al.[50] cover the 25th–75th percentiles. The $E_{abs}$ reported in Cappa et al.[51] were observed in two cities (Fresno and Fontana, California, respectively).

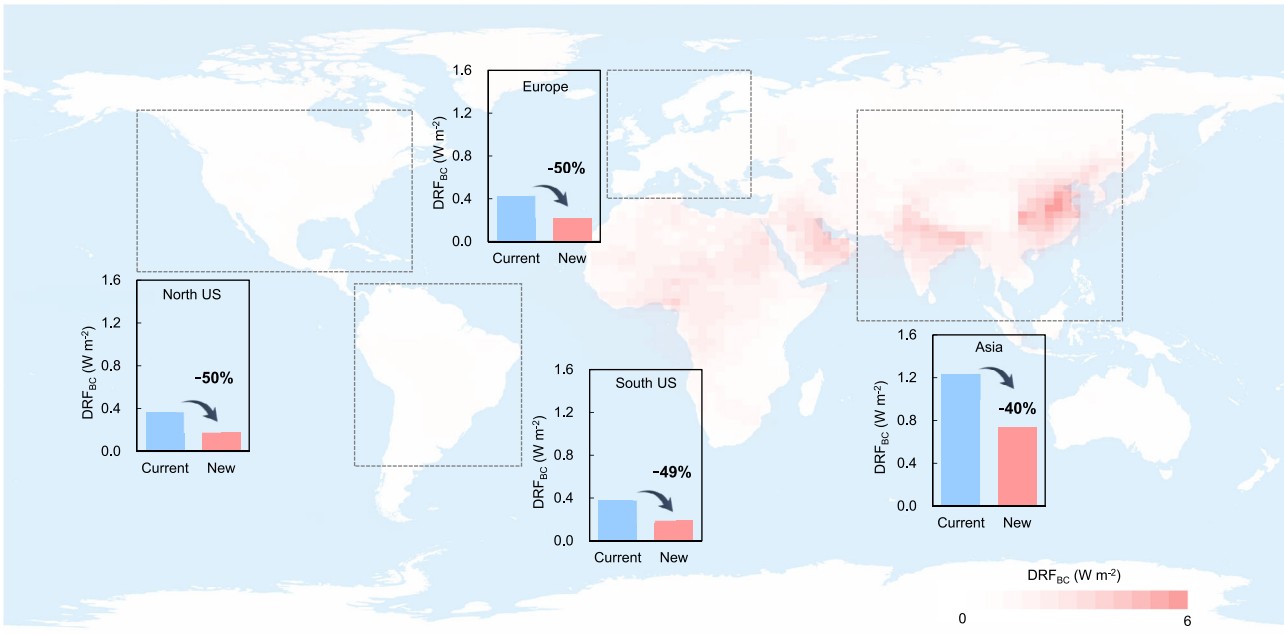

**Fig. 5 | Model simulated black carbon (BC) direct radiative forcing (DRF$_{BC}$) using the new scheme and the conventional scheme in different regions.** The map shows the global distribution of DRF$_{BC}$ using the new scheme. The blue and red bars represent the average simulated DRF$_{BC}$ within each region (gray squares in dash line) using the conventional scheme and the new scheme, respectively.

period was from 26 June to 8 July 2021. The observation locations and periods at other sites are summarized in Table S1.

Besides field observations conducted in this study, observational data from several sites covering different kinds of environment globally was also collected to support our findings. The measurement periods in Shaoguan, Beijing[31], Tokyo[30], Sacramento[9], and the Amazon Tall Tower Observatory (ATTO) were December 2020, November 2014, August 2012, June 2010, and October 2019, respectively. Shaoguan is a regional background site in southeastern China and the Beijing site represents a heavily polluted region in the North China Plain. The Tokyo and Sacramento sites are located in urban areas in Japan and the United States, respectively. The selected observation period at the ATTO site covers a biomass burning episode including some relatively clean days. Detailed information on these observations is summarized in Table S1.

The physical properties of refractory BC particles were measured using single particle soot photometers (SP2, Droplet Measurement Technologies, USA). The operation principle of the SP2 has been well described in previous literature[20,35]. Briefly, sampled particles pass through a 1064 nm Nd:YAG laser beam. BC-containing particles heat up to their boiling point and incandesce. The BC mass can be computed based on its proportional relationship with the peak intensity of the incandescence signal and the BC mass equivalent diameter can be calculated with the known density of BC (normally assumed to be 1.8 g cm$^{-3}$ [36]). The scattering calibration was performed using polystyrene latex spheres (PSL). The rBC mass was calibrated using fullerene soot with known diameter, which was selected by a differential mobility analyzer (DMA, TSI Inc., USA) and its mass was calculated using effective density values presented by Gysel et al.[37]. The leading-edge-only (LEO) fit method developed by Gao et al.[20] was adopted to calculate the scattering cross-sections of BC-containing particles and saturated scattering particles. Therefore, the optical diameters of BC-containing particles can be further determined based on core-shell Mie theory.

## Model simulations

There are two typical methods that are extensively applied for optical calculations in global climate models and regional transport models, i.e., the volume-mixing and core-shell Mie methods. The volume-mixing Mie algorithm assumes that all components are mixed in all individual particles and the mean refractive index is calculated as the volume-weighted average of the refractive indices of each species. The core-shell Mie method assumes that BC is in the center, and other components are coated on the BC core. The shell refractive indices are assumed to be the volume-weighted average of the refractive indices of dissolved components. The volume-mixing Mie algorithm is included in CESM-CAM6. Both volume-mixing and core-shell Mie methods can be used to estimate aerosol optical properties in WRF-Chem. The refractive indices for shortwave radiation and densities of aerosol species in CESM-CAM6 and WRF-Chem model are summarized in Table S3. $E_{abs}$ is the ratio of $MAC_{internal}/MAC_{external}$, where MAC stands for the mass absorption cross section of BC. Since there is no external mixing module in CESM-CAM6 and WRF-Chem, the estimation of $MAC_{external}$ is based on off-line calculation. The default refractive indices and densities in each model are adopted. The BC diameter is assumed to follow a lognormal distribution with a count median diameter of 70 nm[2].

The new mixing state and optical scheme based on our theory frame is established and applied in both CESM-CAM6 and WRF-Chem, which cover different model types. CESM-CAM6 and WRF-Chem are examples of global climate models and regional transport models, respectively. Moreover, CESM-CAM6 uses a modal aerosol module and WRF-Chem uses a sectional aerosol module, which are the two most widely implemented modules in atmospheric models. In the new

scheme, a monodisperse coating thickness of 70 nm derived from $k = 0.014$ is adopted. In CESM-CAM6 and WRF-Chem, the BC core diameter is assumed to follow a lognormal distribution with a count median diameter of 70 nm.

We used the Community Atmosphere Model version 2.1.3 of the Community Earth System Model version 6 (CESM2.1.3-CAM6)[38] in the simulation of light absorption by BC and the global DRF with Modal Aerosol Module 4 (MAM4)[39]. MAM4 includes six aerosol components (BC, sea salt, sulfate, POA, SOA, and dust), which are divided into four modes (primary carbon mode, Aitken mode, accumulation mode and coarse mode), and simulates the mass mixing ratios of six components within each mode. The spatial resolution in the global simulation is 1.9° × 2.5° for a latitude and longitude grid with 70 vertical layers (from 50 m to ~140 km). The simulation is performed for four years (2012–2015) with the spin-up in the first three years and analysis in the last year. The radiative transfer module in the shortwave is calculated by the radiation code RRTMG. The diagnostic calculation of CESM-CAM6 is conducted for the radiative properties of one specific component, namely by double running cases with and without that component. The aerosols in the accumulation mode in this study are resolved with a sectional representation (30 size bins) in the optical calculations.

The BC-induced direct radiative forcing (DRF$_{BC}$) in the conventional models is simulated using the default setting. The DRF$_{BC}$ determined with our new module by using $k$ is performed assuming DRF$_{BC}$ is linear with MAC[2,40]. Thus, the DRF$_{BC\_k}$ can be estimated from the change of MAC$_k$ and MAC in CESM.

WRF-Chem version 3.7 (Weather Research and Forecasting model coupled with Chemistry) was employed in this study, which is an online-coupled meteorology and chemistry model considering multiple physical and chemical processes, including emission and deposition of pollutants, advection and diffusion, gaseous[41] and aqueous chemical transformations, aerosol chemistry, and dynamics[42]. The model has been incorporated in several studies concerning the estimation of aerosol optical properties and its radiative forcing[43,44]. The model domain is centered at 35.0°N, 110.0°E with a grid resolution of 20 km that covers eastern China and the surrounding regions. A total of 30 vertical layers extending from the surface to 50 hPa are utilized in the model. The simulation is conducted for the first two weeks of April 2020, each run covers 36 h, and the last 24 h of output were kept for further analysis. The initial and boundary conditions of meteorological fields were updated from the 6-h NCEP (National Centers for Environment Prediction) global final analysis (FNL) data with 1°×1° spatial resolution. The Rapid Radiative Transfer Model shortwave and long-wave radiation scheme (RRTMG) represents the radiation transfer within the atmosphere[45]. Anthropogenic emissions from power plants, residential combustion, industrial processes, on-road mobile sources, and agricultural activities were derived from the MIX Asian emission inventory database[46]. Emissions of major pollutants such as carbon monoxide, sulfur dioxide, nitrogen oxides, ammonia, and speciated VOCs are all included. The MEGAN (Model of Emissions of Gases and Aerosols from Nature, version2) model embedded in WRF-Chem is used to calculate biogenic emissions online. Soil derived dust emissions are characterized by the GOCART emission schemes. For numerical representation of atmospheric chemistry, we used the CBMZ (Carbon-Bond Mechanism version Z) photochemical mechanism combined with the MOSAIC (Model for Simulating Aerosol Interactions and Chemistry) aerosol model[47]. Major aerosol components include BC, organic mass, sulfate, nitrate, ammonium, and other inorganic species. All aerosols were assumed to be spherical particles. The size distribution was divided into four discrete size bins defined by their lower and upper dry particle diameters (0.039–0.156, 0.156–0.625, 0.625–2.5, 2.5–10.0 μm). In each bin, aerosols were assumed to be internally mixed.

## Data availability

The observation and simulation data generated in this study have been deposited in the Figshare database [https://doi.org/10.6084/m9.figshare.22490959]. Data collected from published papers are mentioned in the main text or the SI with corresponding references. Additional data related to this paper may be requested from the authors.

## Code availability

The CESM2 source code can be downloaded from the CESM official website: http://www2.cesm.ucar.edu. The WRF-Chem source code can be downloaded from the WRF-Chem official website: https://www2.mmm.ucar.edu/wrf/users/download/get_source.html. Code related to this paper is available at https://doi.org/10.6084/m9.figshare.22455511. Additional code may be requested from the authors.

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

## Acknowledgements

This study was supported by the National Natural Science Foundation of China 42005082 (Jiaping Wang), 42075098 (Jiandong Wang), The Second Tibetan Plateau Scientific Expedition and Research (STEP) program 2019QZKK0106 (A.D.), and National Key R&D Program of China (2022YFC3701000, Task 5, Jiandong Wang). Jiaping Wang acknowledges support from Jiangsu Provincial Collaborative Innovation Center of Climate Change. R.A.Z. acknowledges support from the Office of Science of the U.S. DOE through the Atmospheric System Research (ASR) program at Pacific Northwest National Laboratory (PNNL). PNNL is operated for DOE by Battelle Memorial Institute under contract DE-AC06-76RLO 1830. Funding for data collection during the CARES field campaign was provided by the Atmospheric Radiation Measurement (ARM) Climate Research Facility, a U.S. DOE Office of Science user facility sponsored by the Office of Biological and Environmental Research (OBER). S.W. acknowledges support from the Tencent Foundation through the XPLORER PRIZE and the Samsung Advanced Institute of Technology.

## Author contributions

Jiandong Wang, Jiaping Wang, and A.D. conceived and performed the research; R.C. supported theoretical analysis; Jinbo Wang, X.C., B.H., N. Moteki, R.A.Z., P.C., N. Ma, Y.Z., and Q.W. provided observational data; Y.J., C.L., G.C., and Z.W. supported the model simulation; Jiaping Wang and Jiandong Wang wrote the manuscript; M.O.A. and A.D. discussed the results and revised the manuscript; J.J., W.N., X.H., J.C., R.A.Z., Y.C., H.S., J.X., T.L., X.Q., S.W., and J.H. commented on the manuscript.

## Competing interests

The authors declare no competing interests.
