## [Peer Review File · Nature Communications]

Unified theoretical framework for black carbon mixing state allows greater accuracy of climate effect estimationReviewer #1 (Remarks to the Author):

Review of "Unified theoretical framework for mixing state of black carbon allows more accurate estimation of its climate effect" by Wang et al.

Aerosols, particularly the radiatively important compounds like black carbon (BC) play important roles in the climate system. Mixing state is a complex property of BC and its description in numerical models is quite challenging. Most previous studies focused on the heterogeneity of the black carbon (BC) mixing state, while this study found a uniform framework in the mixing state of BC. In this manuscript, the authors utilized both theoretical derivation and model simulations to characterize mixing state and got a better estimation of BC's radiative forcing. They have found the self-similarity of BC mixing state, which greatly simplify its characterization in model calculation from both accuracy and computational efficiency.

The study is well designed and nicely presented. To my knowledge, this novel work is potentially a milestone for aerosol simulations in the climate/earth system models, and the publication of this article will attract widespread attention and have a far-reaching impact in the community. I would like to suggest a couple of revisions before publication.

- 1) Eq.1: In the theoretical derivation, this study considers growth and sink as the main processes influencing BC mixing states. Besides the both processes, have you also considered the impacts of emission of BC? How does it affect BC size distributions? In another word, does the emission process affect the slope (S/GR)? In addition, the growth rate is determined by condensation and coagulation, which should have diurnal cycles associated with solar radiation and air temperature, etc. Does these affect the assumption of steady states?**
- 2) It is impressive that one parameter k is good enough for describing mixing state. Fig. S3 and Fig. S4 are the main evidences for this derivation. Please provide the detailed configuration of optical calculation in Fig. S3 and Fig. S4.**
- 3) Line 79, does the "concentration" represent "mass concentration"? If so, please specify.**
- 4) Line 84, what are the definitions of $n(D_c)$ and $n(D_p)$?**
- 5) Line 2 in SI, it would be better to explain more about the D_p space.**
- 6) Please give a definition of E_{abs} in Page 3, Line 122 or in the method part.**
- 7) For the four D_c bins, should it be "110-119 nm, 120-129 nm, 130-139 nm, and 140-150 nm"?**
- 8) For model simulation, please give some references for CESM and MAM4.**
- 9) Page 3, please check these paragraphs and make sure that all the symbols of variables are in *Italic*.**
- 10) In Fig.3, a shared legend or a description in the figure caption can be used since they are same in all subplots.**

Reviewer #2 (Remarks to the Author):

In this manuscript, the authors have tried to develop a theoretical framework to describe the mixing state that seems can be universally utilized. The authors demonstrated that the size distribution of BC-containing particles is independent of BC core size, based on which a new mixing state module is established and applied in global and regional models. Such application in climate models show a smaller estimation of radiative effects of BC, which is extensively acknowledged to be overpredicted in current models. Generally speaking, the manuscript is well written and to the point. However, I do have several concerns/suggestions as listed below.

Major Comments:

- 1. One of my main suggestions is to extend theoretical derivation in the main text. In my opinion, the main novelties of this work are how to get a universally-applied theoretical**

formula considering the complex processes in the atmosphere. I do not mean that all the derivation needs to be elaborated in the main text; however, some key information, at least, ought to be presented in the Results section rather than in the Supplementary Materials.

2. The uncertainties of the method could be considered. It is claimed that the parameters, such as regression yields k , have a wide range. While such method and parameters are directly used in the climate model, how about the uncertainties in absorbing efficiency of BC-containing particles? Do the uncertainties differ a lot in different regions or under different conditions, like freshly-emitted particles and long-range transported ones? More evaluation of the uncertainty while using this method is suggested to be discussed.

Technical points:

1. What did the authors mean by 'scavenging'? Generally, scavenging is associated with cloud or precipitation processes. But in the atmosphere, dry deposition might be of more importance in shaping BC size distribution. If so, "deposition" could be better than "Scavenging".

2. Line 68-69, "bridges the gap between observations and model simulations". The expression of this phrase is strange and it is recommended to modify.

3. Line 78: the particle diameter distributions of what? BC or BC-containing aerosol? Needs to be specified.

4. Line 85: please define $n()$

5. Line 85-86, the sentence needs to be rewritten.

6. Line 127: please delete "exhibited in"

7. Line 103: The linear regression yields the slopes' \diamond The linear regression yields of slopes

Response to the comments of Reviewer #1

Reviewer #1: Aerosols, particularly the radiatively important compounds like black carbon (BC) play important roles in the climate system. Mixing state is a complex property of BC and its description in numerical models is quite challenging. Most previous studies focused on the heterogeneity of the black carbon (BC) mixing state, while this study found a uniform framework in the mixing state of BC. In this manuscript, the authors utilized both theoretical derivation and model simulations to characterize mixing state and got a better estimation of BC's radiative forcing. They have found the self-similarity of BC mixing state, which greatly simplify its characterization in model calculation from both accuracy and computational efficiency. The study is well designed and nicely presented. To my knowledge, this novel work is potentially a milestone for aerosol simulations in the climate/earth system models, and the publication of this article will attract widespread attention and have a far-reaching impact in the community. I would like to suggest a couple of revisions before publication.

Response: We are very grateful to the reviewer for the constructive comments. We have implemented all suggestions in the revised manuscript. Please find our point-by-point responses listed below. The reviewer's comments are in *Italic* followed by our responses and revisions (in blue).

1) Eq.1: In the theoretical derivation, this study considers growth and sink as the main processes influencing BC mixing states. Besides the both processes, have you also considered the impacts of emission of BC? How does it affect BC size distributions? In another word, does the emission process affect the slope (S/GR)? In addition, the growth rate is determined by condensation and coagulation, which should have diurnal cycles associated with solar radiation and air temperature, etc. Does these affect the assumption of steady states?

Response: Thanks for your comments. According to the theoretical derivation, emission of BC doesn't affect the slope Dep/GR , where Dep represents the deposition rate (same as the scavenging rate S in the original manuscript), but it affects the intercept of $\ln(n(D_p)) \sim D_p$. The reason is that $n(D_c)$ is the number concentration of BC particles from the initial emission into the atmosphere, while the slope Dep/GR does not contain emission-related variables (as shown in Eq. R1, i.e., Eq. 5 in the revised manuscript).

$$\ln(n(D_p)) = \ln(n(D_c)) - \frac{Dep}{GR} (D_p - D_c) \quad (R1)$$

We have added the corresponding description in the revised manuscript:

“The slope $k = \frac{Dep}{GR}$ is determined by the deposition rate and the growth rate, and the intercept is determined by $n(D_c)$.”

Regarding the question about impacts of diurnal variation of growth rate on the assumptions, the following explanation can be found in the SI (in theoretical derivation section, Page 3). Overall, the steady-state assumption is applicable in the presence of periodical variations (such as diurnal variation) of meteorological conditions.

GR and Dep are assumed to have a periodical variation with cycling time of τ , that is,

$$\int_t^{t+\tau} GR(t)dt = \overline{GR} \cdot \tau \quad (\text{R2})$$

$$\int_t^{t+\tau} Dep(t)dt = \overline{Dep} \cdot \tau \quad (\text{R3})$$

Hence, Eq. 2 and Eq. 4 can be represented as Eq. R4 and Eq. R5, respectively.

$$\begin{aligned} \Delta D_p &= \int_t^{t+\tau} GR(t)dt \\ &= \overline{GR} \cdot \tau \end{aligned} \quad (\text{R4})$$

$$\begin{aligned} \ln\left(\frac{n(t+\tau)}{n(t)}\right) &= \int_t^{t+\tau} -Dep(t)dt \\ &= -\overline{Dep} \cdot \tau \end{aligned} \quad (\text{R5})$$

It can be observed that Eq. R4 and Eq. R5 have similar formats as Eq. 2 and Eq. 4, only with \overline{GR} and \overline{Dep} instead of GR and Dep , and τ instead of t .

Therefore, the assumption of constant GR and Dep (independent of time) is applicable to periods that are integer multiples of τ or periods much longer than τ . When discussing the mixing state of BC, multiday statistics are often adopted to represent its average condition, in which case the above derivation can be used. Moreover, the steady-state assumption is also applicable for us to determine the overall mixing state of BC on a large scale.

2) *It is impressive that one parameter k is good enough for describing mixing state. Fig. S3 and Fig. S4 are the main evidences for this derivation. Please provide the detailed configuration of optical calculation in Fig. S3 and Fig. S4.*

Response: Thanks for your suggestions. In the optical calculation, we used a lognormal distribution of D_c . The geometric standard deviation was set to 1.8. The mean diameter of D_c was 70 nm and the wavelength was 550 nm. The refractive indices (RI) of the BC and scattering components were set to $1.85 + 0.7i$ according to Bond et al. (2006)¹ and $1.53 + 0i$ (Pitchford et al. 2007)², respectively. To derive the response of MAC to ΔD_p (Fig. S3), ΔD_p from 10 nm to 200 nm with 10 nm intervals was adopted to calculate the mass absorption coefficient.

For the optical calculations in Fig. S4, we used the integral method (ΔD_p varied from 1 nm to 1000 nm with 1 nm interval) and the k-value method to calculate absorption coefficients. The D_p size distribution represented as Eq. 5 in the revised manuscript was adopted and k was set to 0.014. The setting of RI, wavelength, and D_c size distribution were same as in Fig. S3.

We have added above description in the Method section (Page 5).

3) *Line 79, does the “concentration” represent “mass concentration”? If so, please specify.*

Response: The “concentration” at Line 79 means “mass concentration”. We have revised it based on your comments.

4) Line 84, what are the definitions of $n(D_c)$ and $n(D_p)$?

Response: $n(D_c)$ and $n(D_p)$ represent the number concentration of BC-containing particles with diameter of D_c and D_p , respectively. We have added the definition of $n(D_c)$ and $n(D_p)$ in the revised manuscript (Page 2 and Page 3).

“We assume a monodisperse aerosol population (consisting of BC cores only) emitted into atmosphere at time zero with diameter D_c and number concentration $n(D_c)$.”

“The number concentration of particles at D_p , i.e., $n(D_p)$, decays due to deposition process...”

5) Line 2 in SI, it would be better to explain more about the D_p space.

Response: Thanks for your suggestion. The D_p -space means the space where D_p is the scale of distance and particle sizes can be represented as points. The growth of particles can be considered as the movement in D_p space. In real space, we typically express particle concentration at a given latitude, longitude, and altitude, while in D_p space, we express number concentration within a given size range. Similar to the real space, there are two ways to track particle size distribution in D_p space: the Lagrangian approach, which tracks the change in size of each individual particle, and the Eulerian approach, which tracks the change of number concentration in bins or grids in D_p space.

We have added the explanation of D_p space in revised SI.

“Here the D_p -space means the space where D_p is the scale of distance and particle sizes can be represented as points. The growth of particles can be considered as the movement in D_p space.”

6) Please give a definition of E_{abs} in Page 3, Line 122 or in the method part.

Response: Thanks. The absorption enhancement factor, E_{abs} , is the ratio between aerosol absorption coefficient before and after removal of coating, which is a widely used parameter to represent BC light absorption amplification. We have added the definition of E_{abs} in the revised manuscript (Page 4).

“The absorption enhancement factor, E_{abs} , is the ratio between the aerosol absorption coefficients before and after removal of coating, which is a widely used parameter to represent BC light absorption amplification.”

7) For the four D_c bins, should it be “110-119 nm, 120-129 nm, 130-139 nm, and 140-150 nm”?

Response: Thanks for your suggestion. Technically, we prefer to use a left-closed and right-open interval (e.g., $110 \text{ nm} \leq D_c < 120 \text{ nm}$). Taking the intervals 110-120 nm and 120-130 nm as examples, a particle with diameter of 110 nm is classified into the first bin, and aerosol with diameter of 120 nm is classified into the second bin. If it were changed to 110-119 nm, a particle with diameter of 119.5 nm could not be accurately classified to a bin. Therefore, we would like to keep the original form here.

8) For model simulation, please give some references for CESM and MAM4.

Response: Thanks for your suggestion. We have added references for CESM2³ and MAM4⁴.

“We used the Community Atmosphere Model version 2.1.3 of the Community Earth System Model version 6 (CESM2.1.3-CAM6)⁴⁰ in the simulation of light absorption by BC and the global DRF with Modal Aerosol Module 4 (MAM4)⁴¹.”

9) Page 3, please check these paragraphs and make sure that all the symbols of variables are in *Italic*.

Response: Thanks. We have made all symbols *Italic* in revised manuscript based on your suggestion.

10) In Fig.3, a shared legend or a description in the figure caption can be used since they are same in all subplots.

Response: Thanks. We have modified this figure according to your suggestion (shown below).

Fig. 3 BC size distributions with different D_c ranges from SP2 observations at four sites. Circles are BC D_p distributions with four selected D_c ranges. Dashed lines represent the linear regression of each distribution. $n(D_p)$ with each D_c range normalized.

References

1. Bond, T.C., and Bergstrom, R.W. (2006). Light absorption by carbonaceous particles: An investigative review. *Aerosol Science and Technology* 40, 27–67. 10.1080/02786820500421521.
2. Pitchford, M., Malm, W., Schichtel, B., Kumar, N., Lowenthal, D., and Hand, J. (2007). Revised Algorithm for Estimating Light Extinction from IMPROVE Particle Speciation Data. *Journal of the Air & Waste Management Association* 57, 1326–1336. 10.3155/1047-3289.57.11.1326.
3. Danabasoglu, G., Lamarque, J. -F., Bacmeister, J., Bailey, D.A., DuVivier, A.K., Edwards, J., Emmons, L.K., Fasullo, J., Garcia, R., Gettelman, A., et al. (2020). The Community Earth System Model Version 2 (CESM2). *J. Adv. Model. Earth Syst.* 12. 10.1029/2019MS001916.
4. Liu, X., Ma, P.-L., Wang, H., Tilmes, S., Singh, B., Easter, R.C., Ghan, S.J., and Rasch, P.J. (2016). Description and evaluation of a new four-mode version of the Modal Aerosol Module (MAM4) within version 5.3 of the Community Atmosphere Model. *Geosci. Model Dev.* 9, 505–522. 10.5194/gmd-9-505-2016.

Response to the comments of Reviewer #2

Reviewer #2: In this manuscript, the authors have tried to develop a theoretical framework to describe the mixing state that seems can be universally utilized. The authors demonstrated that the size distribution of BC-containing particles is independent of BC core size, based on which a new mixing state module is established and applied in global and regional models. Such application in climate models show a smaller estimation of radiative effects of BC, which is extensively acknowledged to be overpredicted in current models. Generally speaking, the manuscript is well written and to the point. However, I do have several concerns/suggestions as listed below.

Response: We appreciate the reviewer's constructive comments. We have implemented all suggestions for improvement in the revised manuscript. Please find our point-by-point responses listed below. The reviewer's comments are in *Italic* followed by our responses and revisions (in blue).

Major Comments:

1) One of my main suggestions is to extend theoretical derivation in the main text. In my opinion, the main novelties of this work are how to get a universally-applied theoretical formula considering the complex processes in the atmosphere. I do not mean that all the derivation needs to be elaborated in the main text; however, some key information, at least, ought to be presented in the Results section rather than in the Supplementary Materials.

Response: Thanks for your suggestion. We agree that the theoretical derivation is essential to show this self-similarity. We have added Eqs. 1 to 4 and related descriptions to the Result section from the Method section (shown as follows) since they are the key formulas. Other detailed equations and optical calculation formulas are kept in the Method and Supplementary Materials to maintain the readability of this article.

We have added the following paragraphs in the revised manuscript (Page 2):

“We assume a monodisperse aerosol population (consisting of BC cores only) emitted into the atmosphere at time zero with diameter D_c and number concentration $n(D_c)$. After being emitted into the atmosphere, the particles experience both growth and deposition progresses continuously, which form a steady-state²⁹, that is, the size distribution of BC-containing particles is approximately steady (although the overall mass concentration may change). The growth of BC-containing particles via condensation and coagulation results in an enlarged particle size, whose change as a function of time is represented by the growth rate (GR).

$$\frac{d(D_p)}{dt} = GR \quad (1)$$

The time evolution of the diameter of BC-containing particles (D_p) can be integrated to give

$$D_p = GR \cdot t + D_c \quad (2)$$

At the same time, BC particles are removed by deposition process with the rate of Dep . The number concentration of particles at D_p , i.e., $n(D_p)$, decays due to deposition process is

$$dn(D_p) = -Dep \cdot n(D_p) dt \quad (3)$$

Then, the time evolution of $n(D_p)$ can be integrated as

$$n(D_p) = n(D_c) \cdot e^{-Dep \cdot t} \quad (4)$$

Based on the steady-state approximation, Eq. 2 and Eq. 4 can be combined and time term t can be eliminated. Taking the logarithm on both sides, we obtain the following equation:

$$\ln(n(D_p)) = \ln(n(D_c)) - \frac{Dep}{GR} (D_p - D_c) \quad (5)$$

The slope $k = \frac{Dep}{GR}$ is determined by the deposition rate and the growth rate, and the intercept is determined by $n(D_c)$. Note that we adopted a simplified derivation in the above theoretical analysis for better understanding. A more rigorous theoretical derivation as well as the interpretation of the dependency of GR and Dep on D_p and time can be found in the SI.”

2)The uncertainties of the method could be considered. It is claimed that the parameters, such as regression yields k , have a wild range. While such method and parameters are directly used in the climate model, how about the uncertainties in absorbing efficiency of BC-containing particles? Do the uncertainties differ a lot in different regions or under different conditions, like freshly-emitted particles and long-range transported ones? More evaluation of the uncertainty while using this method is suggested to be discussed.

Response: Thanks for your suggestions. Yes, there is a variation of the slope k across regions and conditions. This difference mainly comes from the ratio of growth rate and deposition rate. As shown in Figure 2, the slope k varies from 0.008 to 0.020 in different regions. However, the observational data we collected covers different types of environments including polluted cities (e.g. Beijing site is affected by fresh vehicle emission), suburb, rural area, clean area (e.g. Maqu is a background site with little anthropogenic emissions and is generally affected only by long-range transported particles), and regions influenced by biomass burning. Therefore, we believe that this range of k can represent the property of most environments. When using this method

for optical calculations, we computed the uncertainty range of MAC and E_{abs} by using the upper and lower limit of k . The derived MAC using CESM-CAM6 varies from 7.5 to 9.5 and E_{abs} ranges from 1.3 to 1.6. In WRF-Chem, the simulated MAC varies from 6.5 to 8.6 and E_{abs} ranges from 1.3 to 1.7.

We have added the uncertainty range in the revised manuscript (Page 5).

“Using the new module, the calculated E_{abs} in CESM-CAM6 and WRF-Chem are 1.4 (1.3-1.6) and 1.4 (1.3-1.7), respectively, which agrees well with observation data”

Technical points:

1) *What did the authors mean by 'scavenging'? Generally, scavenging is associated with cloud or precipitation processes. But in the atmosphere, dry deposition might be of more importance in shaping BC size distribution. If so, “deposition” could be better than “Scavenging”.*

Response: Thanks for your suggestion. We have changed “scavenging” to “deposition” in the revised manuscript.

2) *Line 68-69, “bridges the gap between observations and model simulations”. The expression of this phrase is strange and it is recommended to modify.*

Response: Thanks for your suggestion. We have rewritten this sentence in the revised manuscript (Page 1).

“Our study links observations with model simulations in both mixing state and light absorption.”

3) *Line 78: the particle diameter distributions of what? BC or BC-containing aerosol? Needs to be specified.*

Response: Thanks for your comment. Here it should be BC-containing particles. We have clarified this in the revised manuscript (Page 2).

“...that is, the size distribution of BC-containing particles is approximately steady”

4) *Line 85: please define $n()$*

Response: Thanks for your suggestion. $n(D_c)$ and $n(D_p)$ represent the number concentration of BC-containing particles with diameter of D_c and D_p , respectively. We have added the definition of $n(D_c)$ and $n(D_p)$ in revised manuscript (Page 2 and Page 3).

“We assume a monodisperse aerosol population (consisting of BC cores only) emitted into atmosphere at time zero with diameter D_c and number concentration $n(D_c)$.”

“The number concentration of particles at D_p , i.e., $n(D_p)$, decays due to deposition process...”

5) *Line 85-86, the sentence needs to be rewritten.*

Response: Thanks. This sentence has been modified to provide the definition of D_c and D_p in the revised manuscript (shown as follows).

“We assume a monodisperse aerosol population (consisting of BC cores only) emitted into atmosphere at time zero with diameter D_c and number concentration $n(D_c)$.”

“The time evolution of the diameter of BC-containing particles (D_p) can be integrated...”

6) *Line 127: please delete "exhibited in"*

Response: Thanks. We have deleted it in the revised manuscript according to your suggestion.

7) *Line 103: The linear regression yields the slopes' ∅The linear regression yields of slopes*

Response: Thanks for your suggestion. We have rewritten this sentence in the revised manuscript (Page 4).

“The slope k of linear regression, ranging from 0.008 to 0.020, provides a useful parameter to quantify BC size distribution and its absorption enhancement.”